# Assessment of the Health of Soils Contaminated with Ag, Bi, Tl, and Te by the Intensity of Microbiological Activity

**DOI:** 10.3390/life13071592

**Published:** 2023-07-20

**Authors:** Tatiana Minnikova, Sergei Kolesnikov, Dmitry Khoroshaev, Natalia Tsepina, Natalia Evstegneeva, Alena Timoshenko

**Affiliations:** 1Academy of Biology and Biotechnology, Russian Academy of Sciences, D.I. Ivanovsky, Southern Federal University, Rostov-on-Don 344090, Russia; kolesnikov1970@list.ru (S.K.); cepinanatalia@yandex.ru (N.T.); natalja.evstegneewa@yandex.ru (N.E.); aly9215@mail.ru (A.T.); 2Pushchino Scientific Center for Biological Research of the Russian Academy of Sciences, Federal Research Center, Pushchino 142290, Russia; d.khoroshaev@gmail.com

**Keywords:** haplic chernozems, haplic arenosols, haplic cambisols, metabolic coefficient, specific microbial respiration

## Abstract

Soil health is the basis of ecological and food security for humanity. Among the informative indicators of soil health are microbiological indicators based on the intensity of the carbon dioxide release from the soil. The reaction of the microbial community of Haplic Chernozem Loamic, Haplic Arenosols Eutric, and Haplic Cambisols Eutric to contamination with oxides and nitrates of Ag, Bi, Tl, and Te at doses of 0.5, 1, 3, 10, and 30 derived specific permissible concentrations (SPC) was analyzed in the conditions of a vegetation experiment (the exposure period was 10 days). One derived concentration is assumed to be equal to three background concentrations of the element in the soil. The carbon content of microbial biomass in Haplic Chernozem varied between the experimental options from 6 to 218 mg/kg of soil; in Haplic Arenosols, from 3 to 349 mg/kg of soil; and in Haplic Cambisols, from 7 to 294 mg/kg of soil. Microbial biomass was a more sensitive indicator of contamination by the studied pollutants than basal soil respiration. A decrease in specific microbial respiration was found when Haplic Cambisols were contaminated with Ag, Bi, Te, and Tl oxides. Te and Tl nitrates had a significant toxic effect on each type of soil. At the maximum dose of Tl and Te nitrate, a decrease in basal soil respiration of 56–96% relative to the control and an increase in the metabolic coefficient by 4–6 times was found. The toxicity series of heavy metals averaged for all types of soils in terms of microbiological activity was established: Bi > Ag > Te > Tl (oxides) and Te > Tl > Ag > Bi (nitrates). Nitrates of the elements were more toxic than oxides. Soil toxicity due to Ag, Bi, Tl, and Te contamination was dependent on soil particle size distribution, organic matter content, and soil structure. A series of soil sensitivity to changes in microbial biomass and basal soil respiration when contaminated with the studied pollutants: Haplic Arenosols > Haplic Chernozems > Haplic Cambisols. When diagnosing and assessing the health of soils contaminated with Ag, Bi, Tl, and Te, it is advisable to use indicators of soil microbiological activity.

## 1. Introduction

Pollution with heavy metals and metalloids makes a significant contribution to the degradation of soil cover and the reduction of crop yields and quality worldwide. They belong to the priority pollutants. However, not all heavy metals and metalloids have been studied equally well. Environmental pollution with Hg, Pb, Cd, Cu, Zn, Cr, Ni, Mo, Co, F, B, Se, As, Mn, Ba, V, Sb, Sr, Sn, and W is more common and has been studied better [1,2,3,4,5,6,7,8,9,10,11,12]. However, the ecological consequences of soil pollution with metals and metalloids such as Ag, Bi, Te, Tl, and other “rare” heavy metals have not been studied well. At the same time, the scale and degree of soil pollution by them are increasing every year. Thousands of tons of organic and mineral raw materials contaminated with heavy metals regularly enter the environment [13].

The main sources of Ag, Bi, Te, and Tl pollution include coal combustion plants at thermal power plants, oil refineries, and cement production [14,15,16]. In contaminated soils, Ag concentrations vary from 8.0 to 35.9 mg/kg [17,18,19,20], and Bi from 0.15 to 1891 mg/kg [21,22]. The Te content is about 0.166 mg/kg [23], 11 mg/kg [14], 290 mg/kg near ore deposits [24], Tl—up to 7 mg/kg [25,26], and 19 mg/kg [27].

Studies of the ecological state of soils after anthropogenic impacts on a wide range of enzymes are rare [28,29,30,31,32,33,34,35]. There are studies confirming the toxic effect of Te and Tl on the activity of enzymes in various living systems [36,37,38,39,40]. However, unlike Ag, the study of the influence of Bi, Te, and Tl on the enzymatic activity of soils is given limited attention, despite the relevance of this problem. There are only a few studies on the effect of Bi, Te, and Tl on the enzymatic activity of soils [41,42,43,44].

The intensity of the basal (microbial) respiration of the soil, along with the amount of microbial biomass, is often used to characterize soil microbocenoses in different ecosystems and their response to different types of agricultural land cultivation and land use types [45,46,47,48,49,50,51]. In addition, these indicators can serve as indicators of soil condition under various types of anthropogenic impact [52,53,54,55,56]. The substrate-induced respiration (SIR) method makes it possible to evaluate microbial biomass based on the physiological response of soil microbiocenosis to the introduction of an easily accessible nutrient substrate, usually glucose [42,43,44,45,46,47,48,49,50,51,52,53,54,55,56,57,58,59]). Simultaneous determination of the basal respiration of soil and microbial biomass by the SIR method makes it possible to calculate ecophysiological indicators of microbial communities, such as the metabolic coefficient (qR) and the specific respiration rate of microbiocenosis (V_SIR_). By varying the magnitude of these indicators, it is possible to draw conclusions about the state of microbial communities, which can be useful for assessing adverse anthropogenic impacts.

Microbiological indicators of the soil, including basal respiration, are widely used as indicators of the toxic effects of heavy metals and metalloids in contaminated soils [60]). An increase in the concentration of heavy metals in the soil leads to disruption of the functioning of microbial communities, which can lead to both a decrease and an increase in the heterotrophic activity of the soil [61,62,63,64,65,66]. These changes are reflected in changes in the size of microbial biomass and the intensity of its respiration [67]. However, the reaction of basal soil respiration and the amount of microbial biomass to pollutants depends on the nature of the pollutant and the properties of the soil [68,69,70,71].

The impact of soil pollution with such rare heavy metals as silver (Ag), bismuth (Bi), thallium (Tl), and tellurium (Te) on the biological properties of soils has been studied differently. The effect of silver and its compounds (including nanoparticles) on biota and plants has been studied to a greater extent [18,19,42,72], compared with contamination with bismuth [73,74,75], tellurium [41,75,76], and thallium [75,77,78].

The work objective is to assess the intensity of microbiological activity in soils contaminated by Ag, Bi, Tl, and Te.

## 2. Materials and Methods

### 2.1. Soils

To set up a vegetation simulation experiment, soils differing in mechanical composition, organic matter content, and pH value were used. The first object of the study was Haplic Chernozems Loamic soil [79], selected on the arable land of the Botanical Garden of the Southern Federal University, Rostov-on-Don (47°14′17.54″ N; 39°38′33.22″ E). The second object of the study was Haplic Arenosols Eutric soil [79]. Soil samples were taken on arable land in the Ust’-Donetsk district of the Rostov Region (47°46.015″ N; 40°51.700″ E). The third object of the study was Haplic Cambisols Eutric soil [79], selected in the beech-hornbeam forest near the village Nickel, Republic of Adygea (44°10.649″ N; 40°9.469″ E). Soil samples were taken from the topsoil of the humus-accumulative horizon (A—0–10 cm) in May–July 2022.

### 2.2. Heavy Metals

Oxides and nitrates of Ag, Bi, Te, and Tl were used to pollute the soil. Silver was introduced into the soil in the form of an oxide (Ag_2_O, Alfa Alesar, Haverhill, MA, USA) and silver nitrate solution (AgNO_3_, Sigma-Aldrich, St. Louis, MO, USA). Bismuth was introduced into the soil in the form of oxide (Bi_2_O_3_, Sigma-Aldrich, USA) and a solution of bismuth nitrate (Bi(NO_3_)_3_, Sigma-Aldrich, USA). Tellurium was introduced into the soil in the form of oxide (TeO_2_, Sigma-Aldrich, USA) and a solution of tellurium nitrate (Te_2_O_3_(OH)NO_3_, Sigma-Aldrich, USA). Thallium was introduced into the soil in the form of oxide (Tl_2_O_3_, Sigma-Aldrich, USA) and a solution of thallium nitrate (Tl(NO_3_)_3_, Sigma-Aldrich, USA). The compounds were introduced into the soil in terms of doses of 0.5, 1, 3, 10, and 30 SPCs (Specific Permissible Concentrations). One SPC is equal to three background concentrations of the element in the soil, since for many heavy metals, their toxicity manifests from this concentration. The calculation of the SPC is presented in Table 1.

One SPC is assumed to be equal to three background concentrations of the element in the soil since for many heavy metals, their toxicity manifests from this concentration [80].

### 2.3. Simulation Experiment

For the simulation experiment, soil samples were dried to an air-dry consistency. Three hundred grams of dried and sifted soil were placed in each vegetative vessel; each option was laid in a 3-fold repetition. Oxides and nitrates of Ag, Bi, Tl, and Te were introduced into the soil in terms of 0.5, 1, 3, 10, and 30 SPCs. The metal oxides (in terms of metal) were added in dry form and thoroughly mixed with the soil, and nitrate samples were introduced in the form of an aqueous solution. The introduction of oxides into the soil is due to the fact that it is in this form that heavy metal compounds are mainly present in the soil. Soil contaminated with oxides and nitrates of Ag, Bi, Tl, and Te was incubated in laboratory conditions for 10 days at a temperature of 22 °C and a soil humidity of 25%.

The content of organic carbon (***C_org_***) in the soil was determined photometrically—by the amount of the Cr^3+^ ion formed according to the method of Tjurin in the modification by Nikitin [81,82,83].

### 2.4. Determination of the Basal Respiration of Soils and Microbial Biomass

Before the analysis, soil samples were brought to an air-dry state and sifted through a sieve with a diameter of 2 mm. The field water capacity (FWC) of bulk samples was determined by the method of tubes [84].

The rate of microbial respiration (basal respiration, ***V_basal_***) in the studied soil samples was determined in laboratory conditions by the intensity of CO_2_ release. To do this, 10–12 g of air-dry soil from the averaged sample was placed in glass vials with a volume of 120 mL in a 3-fold repetition. The soil was moistened to 60–80% FWC, after which it was pre-incubated for 8–10 days at a temperature of 22 °C. To prevent the samples from drying out, the vials were covered with a film, which generally did not interfere with gas exchange but limited moisture evaporation. Before determining the rate of CO_2_ release, the vials were ventilated, hermetically sealed with rubber stoppers, and then incubated at a temperature of 22 °C for 17–23 h. The concentration of CO_2_ in the vial was determined using an infrared gas analyzer Li-820 (LiCor, Lincoln, NE, USA). The calculation of the microbial respiration rate (µg C/g/h) was carried out according to Equation (1):(1)$$V_{basal}=\frac{\left(C_{1}-C_{0}\right)\times V\times M\left[C\right]\times 1000}{m\times V_{m}\times t\times 100}$$
where *C*_0_ and *C*_1_—the initial and final concentrations of CO_2_ in the vial (%), *V*—volume (mL), *M*[*C*]—carbon molar mass (12 g/mol), *m*—weight of absolutely dry soil (g), *V_m_*—molar volume under standard conditions (22.4 L/mol), and *t*—incubation time (h).

The carbon of microbial biomass (***C_mic_***) was determined by substrate-induced respiration [58]. The method assumes the same respiratory response of soil microorganisms in the first hours after the introduction of an easily accessible nutrient substrate (glucose), the value of which reflects the total value of the living active microbial biomass. Glucose was introduced after the determination of basal respiration (***V_basal_***). A glucose solution at a rate of 10 mg of glucose per 1 g of soil was carefully distributed over the soil surface. The final soil moisture after glucose application was 90% FWC. Then, 1 h after the introduction of glucose, the vials were ventilated, closed, and left to incubate at a temperature of 22 °C for 1.5–2.2 h. The rate of substrate-induced respiration (***V_SIR_***) was determined similarly to *V_basal_*. The value of microbial biomass (µg C/g of soil) in terms of carbon was calculated using Equation (2) [58]:(2)$$C_{mic}=\frac{\left(40.04\times V_{SIR}^{*}+0.37\right)\times 1000}{100}$$
where ***V_SIR_***—rate of substrate-induced respiration (mL CO_2_/100 g of soil/h).

To determine the functional state of the microbial community, the metabolic coefficient (qCO_2_) was used, which was calculated as the ratio of substrate-induced and basal respiration rates ***V_basal_/V_SIR_*** [85]), and the specific respiration rate of microorganisms (qR) ***V_basal_/C_mic_*** [86]. The ratio ***C_mic_/C_org_*** is used as an indicator of the equilibrium state of the substance in the soil, which is observed at ***C_mic_/C_org_*** within 2–4%. The higher this index, the more organic matter is fixed in the microbial mass.

### 2.5. Statistical Processing

Statistical data processing was carried out using the Statistica 12.0 package. Statistical data (average values, variance) were determined, and the reliability of various samples was established using variance analysis (Student’s *t*-test) and confidence interval (95%).

## 3. Results

### 3.1. Basal Respiration of Soils

In Haplic Arenosols soil and Haplic Cambisols soil, inhibition was detected after contamination with 30 SPCs of silver oxide by 45 and 20% compared to the control, respectively (Figure 1).

Thallium oxide and nitrate had no effect on the basal respiration of Haplic Chernozem soil but inhibited basal respiration in Haplic Arenosols soil and Haplic Cambisols soil at 30 SPCs of thallium nitrate by 96 and 56% of the control, respectively.

Tellurium oxide and nitrate had no effect on the basal respiration of Haplic Chernozem soil but inhibited basal respiration in Haplic Arenosols soil and Haplic Cambisols soil at 30 SPCs of tellurium nitrate by 96 and 93% of the control, respectively.

Microbial biomass. Silver oxide inhibited the microbial biomass of Haplic Chernozem soil at doses of 3, 10, and 30 SPCs by 11, 26, and 43% of the control (Figure 2). At the same time, silver nitrate was less toxic in Haplic Chernozem soil than oxide, since biomass inhibition was observed at doses of 10 and 30 SPCs—5 and 17% of the control. Microbial biomass in Haplic Arenosols soil when contaminated with silver oxide and nitrate decreased at the maximum dose (30 SPCs) by 89 and 29% of the control, respectively. In Haplic Cambisols soil, a decrease in microbial biomass was found when silver nitrate was contaminated with 3, 10, and 30 SPCs by 11, 12, and 17% of the control.

The microbial biomass of Haplic Cambisols soil when contaminated with bismuth nitrate in 3 and 10 SPCs was inhibited by 18 and 15% of the control, respectively. At the same time, reaching a dose of 30 SPCs returned the content of the microbial biomass of Haplic Cambisols soil to the control level.

Thallium nitrate inhibited the microbial biomass of Haplic Chernozem soil at 10 and 30 SPCs by 55 and 76%, respectively. Microbial biomass in Haplic Arenosols soil at doses of thallium nitrate 3, 10, and 30 SPCs was 13, 83, and 100% lower than in the control. In Haplic Cambisols soil, thallium oxide both increased microbial biomass (3 SPCs) and contributed to its decrease at doses of 0.5, 1, 10, and 30 SPCs by 8–15% of the control, respectively. Thallium nitrate inhibited the microbial biomass of Haplic Cambisols by 20% compared to the control at a dose of 0.5 SPC. With a further increase from 1 to 30 SPCs, the inhibition of Haplic Cambisols soil biomass was 40–67% of the control, respectively.

Tellurium oxide did not have a significant effect on the microbial biomass of Haplic Chernozem soil but stimulated the microbial biomass of Haplic Arenosols soil by 5–8%, showing the effect of hormesis (stimulation of soil biological activity at low doses of pollution). Tellurium nitrate inhibited the microbial biomass of Haplic Chernozem soil at doses of 3, 10, and 30 SPCs by 26, 27, and 99% of the control, respectively. In Haplic Arenosols soil, inhibition of microbial biomass was established with a dose of 0.5 SPC by 10% relative to the control. With an increase in the dose from 1 to 30 SPCs, the microbial biomass of Haplic Arenosols soil decreased by 34–100%, respectively.

Thus, it was found that when measuring the basal respiration of soils, significant differences from the control were revealed at a maximum dose of 30 SPCs, while microbial biomass was affected by contamination with oxides and nitrates at a minimum dose of 0.5 SPC. Consequently, microbial biomass is a more sensitive biological indicator than the basal respiration of soils.

Changes in microbial respiration coefficients, the proportion of carbon in microbial biomass, and the microbial metabolic coefficient revealed that the metabolic coefficient (qCO_2_), the carbon fraction of microbial biomass (***C_mic_/C_org_***), and the coefficient of microbial respiration (qR), in cases of contamination with oxides and nitrates, differed significantly (Table 2).

The share of microbial biomass (***C_mic_/C_org_***) is directly related to the content of organic carbon in the soil. Such a trend of basal respiration in slightly humic soil has been noted previously by Terehova (2021) [87]. In Haplic Chernozem soil, the ratio of ***C_mic_/C_org_*** differed most from the control when contaminated with silver oxide and tellurium nitrate, by 43 and 35%, which was lower than in the control (Table 1).

The smallest changes in the ratio of ***C_mic_/C_org_*** compounds in Haplic Chernozem soil were observed when contaminated with tellurium oxide and silver nitrate and were 12 and 9% below the control, respectively. In Haplic Cambisols soil, the ratio of *C_mic_/C_org_* in comparison with the control was 52 and 69% lower when contaminated with thallium and tellurium nitrate than in the control. Less reduction of ***C_mic_/C_org_*** compared to the control was found when Haplic Cambisols soil was contaminated with bismuth nitrate—16% lower than the control. In Haplic Arenosols soil, when the oxides of Ag, Bi, Te, and Tl and nitrates of Ag and Bi were introduced into the soil, the stimulation of the carbon fraction of microbial biomass (***C_mic_/C_org_***) was 23–69% higher than in the control.

### 3.2. Metabolic Coefficient (qCO_2_)

According to Blagodatskaja et al. (1995), the qR value in the range of 0.1–0.2 indicates a favorable state of the soil microbial community [85]. When qR < 0.1, there is a lack of nutrients in the soil, and qR values exceeding 0.2–0.3 indicate adverse anthropogenic impacts on the soil. At qR > 1.0, there is intensive decomposition of organic matter and a stability violation in soil biocenoses. The lack of fertilizer elements was established according to this scale (qR < 0.1) in Haplic Chernozem soil and Haplic Arenosols soil, except for 10 and 30 SPCs of thallium and tellurium, where an unfavorable condition (qR > 0.2) was expressed as a result of anthropogenic impact. In Haplic Cambisols soil, when contaminated with oxides and nitrates of Ag, Bi, Te, and Tl, adverse anthropogenic impacts on the soil were established, and when contaminated with nitrate of Te and Tl, an intensive decomposition of organic matter and disruption of the stability of soil biocenoses occurred. The maximum stimulation was observed when Haplic Chernozem soil was contaminated with bismuth and thallium oxide—69 and 48% higher than in the control. The coefficient of microbial respiration (qR) of Haplic Chernozem soil was maximal when contaminated with thallium and tellurium nitrate at high doses (3, 10, and 30 SPCs), by 109 and 299% higher than in the control. The introduction of bismuth nitrate into Haplic Chernozem soil caused a decrease of 15% relative to the control. Bismuth oxide and silver nitrate in Haplic Chernozem soil did not affect qR and did not significantly differ from the control. In Haplic Cambisols soil, as well as in Haplic Chernozem soil, the greatest stimulation of qR was found when contaminated with thallium and tellurium nitrates—48 and 196%, respectively. When Haplic Cambisols soil was contaminated with bismuth oxide and nitrate, qR was the lowest—8 and 13% compared to the control. In Haplic Arenosols soil contaminated with Bi, Te, and Tl oxides and nitrate, Bi did not differ from the control. qR stimulation was achieved when Haplic Arenosols soil was contaminated with Ag oxide, nitrate, and Te nitrate by 104%, 12%, and 383%, respectively. The coefficient of microbial respiration inhibition was established when Haplic Arenosols soil was contaminated with Tl nitrate 123% below control.

With an increase in the dose of Ag, Bi, Te, and Tl, the toxicity of Haplic Cambisols soil by the qCO_2_ coefficient both increased and decreased. The coefficient in Haplic Chernozem soil and Haplic Arenosols soil, with the exception of silver oxide, did not differ from the control. When Haplic Arenosols soil was contaminated with 30 SPCs of silver oxide, the qCO_2_ value was 6 times higher than in the control. In Haplic Chernozem soil, qCO_2_ was maximal when contaminated with silver oxide at 10 and 30 SPCs, 46 and 75% higher than the control, respectively. Haplic Cambisols soil had the highest qCO_2_ value, from 0.71 to 0.86, while Haplic Arenosols soil was characterized by a range of 0.14–0.20 when contaminated with Ag and Bi nitrates, and Haplic Chernozem soil was the least of all: 0.09–0.12. However, tellurium and thallium nitrates had different effects on qCO_2_. When Haplic Arenosols soil was contaminated with 30 SPCs of tellurium and thallium, the stimulation of the metabolic coefficient (qCO_2_) was 38 and 103 times compared to the control. In Haplic Chernozem soil, with contamination of 30 SPCs of tellurium, and with 10 and 30 SPCs of thallium, stimulation of the coefficient (qCO_2_) was found to be 4–6 times compared with the control. In Haplic Cambisols soil, the stimulation coefficient (qCO_2_) at a dose of 30 SPCs of tellurium was 5 times greater than in the control.

### 3.3. Ranking the Toxicity of Elements 

According to the Results of Microbial Biomass and Basal Respiration of Soils, the Series of Soil Toxicity were Compiled.

Toxicity series of metal oxides by basal respiration:

Haplic Chernozem: Bi > Te > Ag > Tl

Haplic Arenosols: Te > Ag = Bi > Tl

Haplic Cambisols: Bi > Ag > Te > Tl

Toxicity series of metal nitrates by basal respiration:

Haplic Chernozem: Tl > Te > Ag > Bi

Haplic Arenosols: Te > Tl > Bi > Ag

Haplic Cambisols: Te > Tl > Bi > Ag

The most toxic effects on the basal respiration of soils of different granulometric compositions were exerted by Bi oxide and Te and Tl nitrates.

Toxicity series of metal oxides by microbial biomass:

Haplic Chernozem: Ag > Tl > Te > Bi

Haplic Arenosols: Ag > Bi = Te > Tl

Haplic Cambisols: Bi > Ag > Te > Tl

Toxicity series of metal nitrates by microbial biomass:

Haplic Chernozem: Te > Tl > Ag > Bi

Haplic Arenosols: Te > Tl > Bi > Ag

Haplic Cambisols: Te > Tl > Ag > Bi

The most toxic effect on the microbial biomass of soils of different granulometric compositions was exerted by Ag oxide and Te and Tl nitrates. The average toxicity range for all types of soils when contaminated with metal oxides was: Bi > Ag > Te > Tl; and when contaminated with metal nitrates: Te > Tl > Ag > Bi. Microbiological indicators were the most depressed when soils of different granulometric compositions were polluted with Ag and Bi oxides and Te and Tl nitrates.

A generalized series of soil resistance to changes in basal respiration and microbial biomass is presented as follows: Haplic Arenosols > Haplic Chernozems > Haplic Cambisols.

Thus, the soil least resistant to Ag, Bi, Te, and Tl pollution is Haplic Cambisols soil, and the most resistant is Haplic Arenosols soil. The stability of Haplic Arenosols soil is due to its physico-chemical and physical properties (low humus content, medium reaction, and granulometric composition) due to which metals do not pass into a fixed form and, as a result, do not have a toxic effect on soil biota such as those more heavily loamy humus soils [42].

To assess the correlation ratio with the metal content, correlation coefficients were calculated (Table 3). It was found that metal oxides of Ag, Bi, Te, and Tl directly affected basal respiration when Haplic Chernozem (r = 0.67) and Haplic Cambisols (r = 0.67) were contaminated with thallium. Only tellurium oxide caused a negative coefficient when contaminated Haplic Chernozems soil: r = −0.78.

Oxides of all metals affected Haplic Arenosols soil contamination with negative coefficients r = −0.91, r = −0.54, r = −0.42, and r = −0.68 for Ag, Bi, Te, and Tl, respectively. Ag nitrate affected the microbial biomass of all soils with a negative correlation: Haplic Chernozem (r = −0.95), Haplic Arenosols (r = −1.00), and Haplic Cambisols (r = −0.87). Bismuth exerted an inverse correlation effect on ***V_basal_*** if Haplic Chernozem (r = −0.59) and Haplic Arenosols (r = −0.52). Tellurium nitrate in Haplic Arenosols and Haplic Cambisols according to ***V_basal_*** correlation coefficient r = −0.97 and r = −0.89, respectively; a negative correlation was established with ***C_mic_***: Haplic Chernozems soil (r = −0.92), Haplic Arenosols (r = −0.81), and Haplic Cambisols (r = −0.54). The basal respiration and microbial biomass of Haplic Arenosols when contaminated with thallium nitrate had an inverse correlation with the metal content: r = −0.97 and r = −0.83, respectively.

## 4. Discussions

Contamination of soils of different granulometric compositions with oxides and nitrates of Ag, Bi, Te, and Tl directly affects the basal respiration and microbial biomass of soils: with an increase in the dose of the contaminant, the response of microbiological indicators increases. Soil toxicity, due to Ag, Bi, Tl, and Te contamination, was dependent on soil particle size distribution, organic matter content, and soil structure. The influence of Ag, Bi, Te, and Tl pollution on microbial biomass is most pronounced in slightly humic soil—Haplic Arenosols. These data are consistent with the results of Terehova et al. (2021) on the assessment of microbiological indicators of highly and slightly humic soils when contaminated with Cu, Zn, and Pb [87]. Haplic Cambisols has an acidic reaction of the medium, which leads to the inefficient use of ***C_org_*** and a decrease in the rate of its mineralization [86,88]. A decrease in the efficiency of the use of organic substrate by microorganisms of Haplic Cambisols or an increase in ***V_basal_*** means that most of the substrate is converted into CO_2_, and a smaller part is involved in the production of microbial biomass [89,90,91]. An increase in the specific microbial respiration of the soil in the presence of pollutants was found [92]. The addition of Cu, Zn, and Pb reduced microbial biomass by an average of 49–57%, basal respiration—23–52%; however, the microbial metabolic coefficient (qCO_2_) increased by an average of 9–46% [87]. Microbial biomass reflects the result of heterotrophic activity, which is indirectly related to the enzymatic activity of soils and directly to the number of bacteria [93,94,95]. In this regard, it is important to have an idea of how these metals affect the enzymatic activity, microbiological parameters, and phytotoxicity of soils. It was previously presented that according to the degree of sensitivity to contamination of Haplic Chernozem Calcic with silver, the most sensitive indicators were catalase activity and the abundance of the Azotobacter bacteria. According to the degree of informativeness to contamination with oxides and nitrates of Ag, Bi, Te, and Tl, the biological indicators of Haplic Chernozem Calcic are the activity of dehydrogenases (r = −0.99) and the abundance of the Azotobacter bacteria (r = −0.99) [42]. From the point of view of their resistance to silver contamination, the studied soils are arranged in the following order: Haplic Chernozem Calcic > Haplic Arenosols Eutric ≥ Haplic Cambisols Eutric.

The greatest sensitivity to Bi contamination was revealed by the number of bacteria and dehydrogenases activity, and the greatest informativeness by the abundance of the Azotobacter sp. bacteria (r = −0.55) and germination of radish seeds (r = −0.68) [74]. The order of resistance of the studied soils to Bi was as follows: Haplic Chernozem > Haplic Arenosols > Haplic Cambisols.

In previous studies on the resistance of Haplic Chernozems to contamination with oxides and nitrates of Ag, Bi, Te, and Tl, it was found that when contaminated with oxides, the most informative indicators were the activity of invertase (Ag), urease (Bi, Tl), and phosphatase (Te); when contaminated with nitrates, the most informative indicators were the activity of phosphatase (Ag) and invertases (Bi, Tl, and Te) [75]. Within 3 months from the moment of soil contamination, when contaminated with oxides of Ag, Bi, Te, and Tl, the most sensitive biological indicator was the length of the wheat roots, whereas when contaminated with nitrates of Ag, Bi, Te, and Tl, the most sensitive biological indicator was the total number of bacteria. The most ecotoxic elements among those studied were Tl and Te, both in the form of oxides and nitrates [96]. These results are consistent with data on the high toxicity of Tl and Te oxides and nitrates on the microbial biomass and basal respiration of Haplic Chernozem, Haplic Arenosols, and Haplic Cambisols.

## 5. Conclusions

Contamination with Ag, Bi, Tl, and Te affects soil health, as estimated by indicators of soil microbiological activity. The toxicity series of heavy metals averaged for all types of soils in terms of microbiological activity was established: Bi > Ag > Te > Tl (oxides) and Te > Tl > Ag > Bi (nitrates). Nitrates of the elements are more toxic than oxides. Soil toxicity due to Ag, Bi, Tl, and Te contamination was dependent on soil particle size distribution, organic matter content, and soil structure. The generalized range of soil sensitivity to changes in basal respiration and microbial biomass is presented as follows: Haplic Arenosols > Haplic Chernozems > Haplic Cambisols. The most sensitive indicators of soil microbiological activity are the microbial respiration coefficient (qR) and metabolic coefficient (qCO_2_). When diagnosing and assessing the health of soils contaminated with Ag, Bi, Tl, and Te, it is advisable to use indicators of soil microbiological activity.

## Figures and Tables

**Figure 1 life-13-01592-f001:**
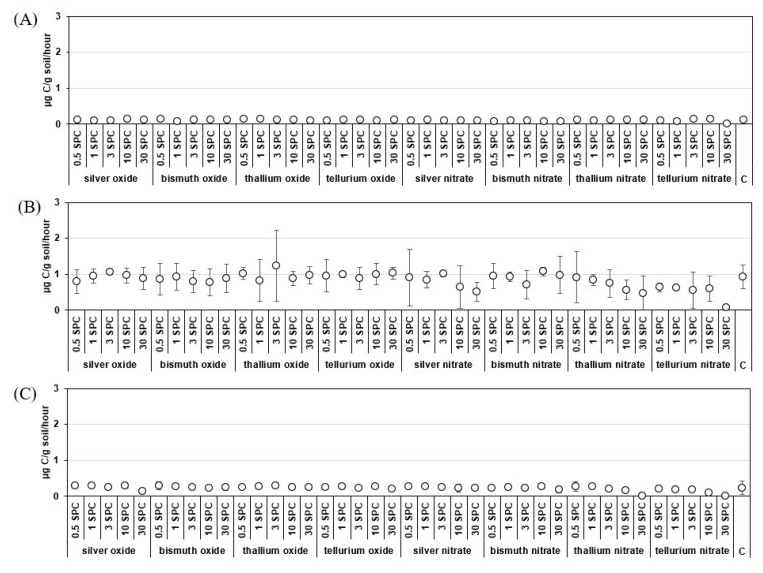
Basal respiration (***V_basal_***) of soils of different granulometric composition contaminated with oxides and nitrates of Ag, Bi, Te, Tl, µg C/g soil/hour: (**A**) Haplic Chernozem (CI = 0.01–0.09); (**B**) Haplic Cambisols (CI = 0.07–0.99); (**C**) Haplic Arenosols (CI = 0.01–0.13). Note: C—control; SPCs—Specific Permissible Concentration; CI—Confidence interval (95%).

**Figure 2 life-13-01592-f002:**
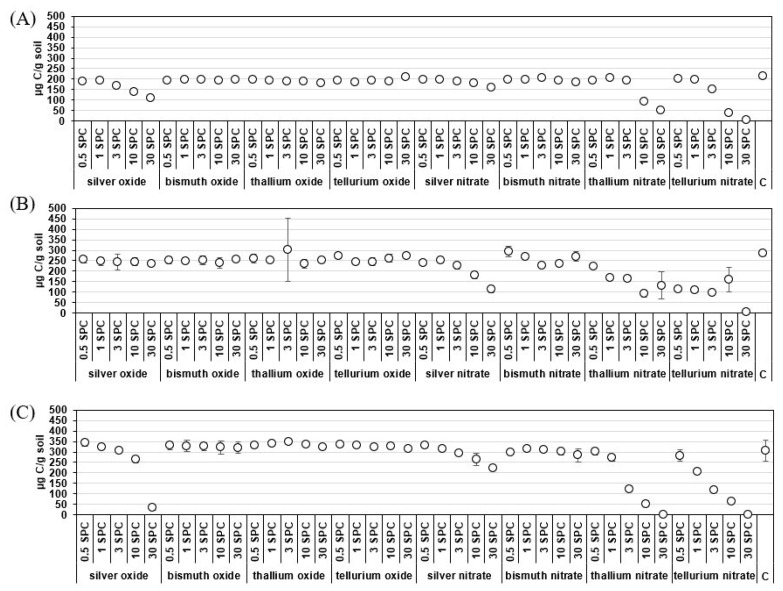
Microbial biomass (C_mic_) of soils of different granulometric composition when contaminated with oxides and nitrates Ag Bi, Te, Tl, µg C/g soil: (**A**) Haplic Chernozem (CI = 0.01–0.29); (**B**) Haplic Cambisols (CI = 0.04–2.00); (**C**) Haplic Arenosols (CI = 0.02–0.43). Note: C—control; SPCs—Specific Permissible Concentration; CI—Confidence interval (95%).

**Table 1 life-13-01592-t001:** Calculation of the specific allowable concentration of heavy metals in soil.

Element	Compound Formula	Background Element Content, mg/kg	Specific Permissible Concentration (SPC), mg/kg
Silver	Ag_2_O	0.10	0.21
	AgNO_3_	0.10	0.16
Bismuth	Bi_2_O_3_	0.27	0.60
	Bi(NO_3_)_3_ × 5H_2_O	0.27	0.62
Thallium	Tl_2_O_3_	0.47	1.05
	TlNO_3_	0.40	0.61
Tellurium	TeO_2_	0.50	0.62
	Te_2_O_3_(OH)NO_3_	0.50	1.49

**Table 2 life-13-01592-t002:** Main indicators of microbiological activity of soils contaminated with oxides and nitrates of Ag, Bi, Te, Tl.

Variants		Haplic Chernozem			Haplic Arenosols			Haplic Cambisols		
		C_miv_/ C_org_	qR	qCO_2_	C_miv_/ C_org_	qR	qCO_2_	C_miv_/ C_org_	qR	qCO_2_
silver oxide	0.5 SPC	0.32	0.05	0.66	1.46	0.06	0.84	0.34	0.24	3.11
	1 SPC	0.34	0.04	0.51	1.35	0.07	0.94	0.31	0.29	3.81
	3 SPC	0.26	0.05	0.59	1.25	0.06	0.84	0.31	0.33	4.34
	10 SPC	0.26	0.07	0.92	1.23	0.09	1.15	0.28	0.30	3.93
	30 SPC	0.18	0.08	1.10	0.16	0.33	3.91	0.33	0.29	3.76
bismuth oxide	0.5 SPC	0.36	0.05	0.67	1.62	0.07	0.91	0.36	0.26	3.46
	1 SPC	0.32	0.03	0.42	1.40	0.06	0.82	0.30	0.29	3.78
	3 SPC	0.34	0.04	0.57	1.39	0.06	0.75	0.31	0.24	3.17
	10 SPC	0.36	0.04	0.56	1.42	0.05	0.72	0.29	0.25	3.26
	30 SPC	0.34	0.04	0.53	1.14	0.06	0.75	0.29	0.26	3.46
thallium oxide	0.5 SPC	0.35	0.06	0.73	1.46	0.06	0.74	0.29	0.30	3.96
	1 SPC	0.35	0.05	0.69	1.19	0.06	0.79	0.30	0.25	3.25
	3 SPC	0.32	0.04	0.59	1.11	0.07	0.86	0.37	0.31	4.10
	10 SPC	0.35	0.05	0.64	1.19	0.06	0.74	0.29	0.28	3.73
	30 SPC	0.25	0.04	0.55	1.11	0.06	0.76	0.29	0.29	3.82
tellurium oxide	0.5 SPC	0.37	0.04	0.51	1.18	0.06	0.75	0.32	0.27	3.51
	1 SPC	0.30	0.05	0.60	1.02	0.06	0.80	0.31	0.31	4.03
	3 SPC	0.44	0.05	0.60	0.94	0.05	0.69	0.29	0.27	3.60
	10 SPC	0.53	0.04	0.56	1.20	0.06	0.81	0.31	0.29	3.82
	30 SPC	0.46	0.04	0.58	1.06	0.05	0.64	0.32	0.29	3.78
silver nitrate	0.5 SPC	0.39	0.04	0.50	1.13	0.06	0.82	0.28	0.29	3.78
	1 SPC	0.37	0.04	0.57	1.11	0.06	0.85	0.30	0.26	3.37
	3 SPC	0.47	0.03	0.45	1.00	0.06	0.86	0.26	0.34	4.44
	10 SPC	0.49	0.04	0.47	0.84	0.06	0.85	0.26	0.27	3.57
	30 SPC	0.44	0.05	0.61	0.98	0.08	1.07	0.16	0.35	4.48
bismuth nitrate	0.5 SPC	0.55	0.03	0.43	1.55	0.06	0.74	0.35	0.24	3.21
	1 SPC	0.50	0.04	0.48	1.56	0.06	0.76	0.36	0.26	3.44
	3 SPC	0.61	0.04	0.47	0.88	0.05	0.70	0.29	0.24	3.15
	10 SPC	0.59	0.03	0.41	0.76	0.07	0.92	0.30	0.35	4.55
	30 SPC	0.57	0.03	0.43	0.94	0.05	0.64	0.37	0.27	3.62
thallium nitrate	0.5 SPC	0.53	0.04	0.55	1.25	0.07	0.89	0.27	0.31	4.09
	1 SPC	0.59	0.04	0.51	0.62	0.07	0.99	0.21	0.38	4.99
	3 SPC	0.47	0.04	0.58	0.38	0.12	1.57	0.20	0.34	4.50
	10 SPC	0.24	0.10	1.34	0.27	0.23	2.84	0.11	0.47	6.05
	30 SPC	0.15	0.19	2.34	0.02	−0.56	2.52	0.16	0.27	3.53
tellurium nitrate	0.5 SPC	0.53	0.04	0.50	1.23	0.06	0.76	0.14	0.43	5.55
	1 SPC	0.53	0.03	0.40	0.47	0.07	0.92	0.13	0.43	5.60
	3 SPC	0.38	0.07	0.98	0.39	0.11	1.49	0.12	0.44	5.69
	10 SPC	0.11	0.28	3.40	0.15	0.12	1.46	0.21	0.28	3.70
	30 SPC	0.02	0.37	1.77	0.01	1.09	2.72	0.01	1.97	11.65
Control		0.48	0.04	0.58	0.82	0.06	0.75	0.40	0.24	3.23

**Table 3 life-13-01592-t003:** Correlation coefficients (r) between content oxide and nitrate of Ag, Bi, Te, Tl with basal respiration (***V_basal_***) and microbial biomass (***C_mic_***).

r	Haplic Chernozem		Haplic Arenosols		Haplic Cambisols	
	*V_basal_*	*C_mic_*	*V_basal_*	*C_mic_*	*V_basal_*	*C_mic_*
oxides						
Ag	0.35	−0.95 **	−0.91 **	−1.00 **	−0.13	−0.87 **
Bi	−0.04	0.49 *	−0.54 *	−0.84 **	0.02	0.34
Te	−0.78 *	−0.88 **	−0.42 *	−0.69 *	−0.14	−0.31
Tl	0.67 *	0.90 **	−0.68 *	−0.85 **	0.64 *	0.52 *
nitrates						
Ag	−0.17	−0.99 **	−0.65 *	−0.95 **	−0.88 **	−0.98 **
Bi	−0.59 *	−0.85 **	−0.52 *	−0.83 **	0.36	0.03
Te	0.53 *	−0.92 **	−0.97 **	−0.81 **	−0.89 **	−0.54 *
Tl	−0.76 *	−0.88 **	−0.97 **	−0.83 **	−0.96 **	−0.75 **

Note: *—*p* < 0.05, **—*p* < 0.001.

## Data Availability

Not applicable.

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
