# Peer review of "Assessment of the Health of Soils Contaminated with Ag, Bi, Tl, and Te by the Intensity of Microbiological Activity"

_life, 2023, doi:10.3390/life13071592_

Round 1
Reviewer 1 Report
I have read the manuscript and I have two notes to add to the text.
Add the effect of studied metals on soil enzymes to the Introduction section. Although their influence is discussed, there is no mention of them in the Introduction.
In the Discussion section, I would add the effect of the studied metals on the availability of carbon and nitrogen in the soil, or the effect on the C/N ratio.
Author Response
Author responses to reviewer (1) comments
(life-2474325)
I have read the manuscript and I have two notes to add to the text.
Add the effect of studied metals on soil enzymes to the Introduction section. Although their influence is discussed, there is no mention of them in the Introduction.
Answer: Information on the effect of metals on the activity of soil enzymes in soils has been added.
In the Discussion section, I would add the effect of the studied metals on the availability of carbon and nitrogen in the soil, or the effect on the C/N ratio.
Answer: These metals are very little studied, so I have not come across articles evaluating the effect of pollution with these heavy metals on the ratio.

Reviewer 2 Report
The manuscript entitled “Assessment of the Health of Soils Contaminated with Ag, Bi, Tl and Te by the Intensity of Microbiological Activity” is devoted to the reaction of the microbial community of three Haplic soils to contamination with Ag, Bi, Tl, and Te, which is important from the point of view of ecology, soil science and bioremediation. The material will be of interest to a wide range of researchers and it is consistent with the aims and scopes of the Life journal.
I have a few fundamental remarks.
1. In the introduction, it is worth indicating the sources of elements entering the soil. They are quite specific, and for a global assessment of the relevance of research, it should be said about the ways of their entry and specific sites where such pollution occurs. In addition, it is necessary to clearly formulate the reasons for the choice of these elements in this work. much attention is paid to the assessment of the use of basal respiration of microorganisms in the management. it is worth shortening this part a little and arranging
2. 74 The work objective is to assess the health of soils I think it's worth reformulating this phrase
3. Are the selected concentrations of elements related to the actual values of these elements in soils under pollution according to literary sources?
4. oxides of most element are poorly soluble, their washout is low
5. The MTT test is used to assess the respiratory activity of the microflora both in solution and on carriers. It is worth using it in comparison with other methods.
6. Figure 1 shows catastrophic error rates for a number of elements. Is it worth it to provide such data?
7. regarding the assessment of the toxicity of elements, it is worth discussing their solubility and the interaction of ions with the soil. Perhaps this is due to the lack of toxicity. I would like to see this in the discussion section. Moreover please explain the differences due to different types of soils.
8. illustrations in the text are quite similar and there are many tables in the text
9. 360 dehydrogenaseы
10. The conclusions should be reconsidered, they should not only state the facts of determining toxicity, but also explain the differences due to different types of soils.
English is worth checking with a native speaker
Author Response
Author responses to reviewer (2) comments
(life-2474325)
The manuscript entitled “Assessment of the Health of Soils Contaminated with Ag, Bi, Tl and Te by the Intensity of Microbiological Activity” is devoted to the reaction of the microbial community of three Haplic soils to contamination with Ag, Bi, Tl, and Te, which is important from the point of view of ecology, soil science and bioremediation. The material will be of interest to a wide range of researchers and it is consistent with the aims and scopes of the Life journal.
I have a few fundamental remarks.
- In the introduction, it is worth indicating the sources of elements entering the soil. They are quite specific, and for a global assessment of the relevance of research, it should be said about the ways of their entry and specific sites where such pollution occurs. In addition, it is necessary to clearly formulate the reasons for the choice of these elements in this work. much attention is paid to the assessment of the use of basal respiration of microorganisms in the management. it is worth shortening this part a little and arranging
Answer: Introduction have been corrected.
- 74 The work objective is to assess the health of soils I think it's worth reformulating this phrase
Answer: The work objective has been corrected.
- Are the selected concentrations of elements related to the actual values of these elements in soils under pollution according to literary sources?
Answer: Yes, they are. One SPC is taken equal to three background concentrations of the element in the soil, since for many heavy metals their toxicity is manifested from this concentration.
Answer: Yes, you are right, oxides are insoluble and do not leach out of the soil.
- The MTT test is used to assess the respiratory activity of the microflora both in solution and on carriers. It is worth using it in comparison with other methods.
Answer: In our experience, we determined whether the microbiological will show using an infrared gas analyzer Li-820. The well-known MTT test method does not give such accuracy as the measurement with an infrared analyzer. Of course, this method is not ideal, which is shown in Figure 1 (Vbasal), where very large errors were found Haplic Cambisols.
- Figure 1 shows catastrophic error rates for a number of elements. Is it worth it to provide such data?
Answer: By presenting such data, we report that the accuracy of determination for different elements varies, including this reported by a large error for Vbasal Haplic Cambisols.
- regarding the assessment of the toxicity of elements, it is worth discussing their solubility and the interaction of ions with the soil. Perhaps this is due to the lack of toxicity. I would like to see this in the discussion section. Moreover please explain the differences due to different types of soils.
Answer: The text of the discussion states that the mobility of heavy metals in soil depends on the content of humus (Corg), pH and particle size distribution.
- illustrations in the text are quite similar and there are many tables in the text
Answer: Yes, fig. 1 and 2 look similar, but this is due to the desire of the authors to compare Basal respiration (Vbasal) and Microbial biomass (Cmic) for three soil types. The figures differ in indicators and units of measurement. The text contains 3 tables that represent the introduced heavy metals and compounds (Table 1), microbiological coefficients that serve as indicators of the state and health of the soil (Table 2) and correlation coefficients (Table 3), reflecting the closeness of the relationship between microbiological activity and the concentration of the element in soil.
- 360 dehydrogenaseы
Answer: Corrected.
- The conclusions should be reconsidered, they should not only state the facts of determining toxicity, but also explain the differences due to different types of soils.
Answer: The conclusion has been corrected

Reviewer 3 Report
The authors should reanalyze the data obtained. Numerous conclusions about the effects of individual compounds on microbial activity are presented, but none of them are supported by statistical analysis. I see a description of statistical analysis in the Materials and Methods section, but in the results I do not see its application. It is not shown which combinations are statistically significantly different.
Author Response
Author responses to reviewer (3) comments
(life-2474325)
The authors should reanalyze the data obtained. Numerous conclusions about the effects of individual compounds on microbial activity are presented, but none of them are supported by statistical analysis. I see a description of statistical analysis in the Materials and Methods section, but in the results I do not see its application. It is not shown which combinations are statistically significantly different.
Авторам следует провести повторный анализ полученных данных. Представлены многочисленные выводы о влиянии отдельных соединений на микробную активность, но ни один из них не подтверждается статистическим анализом. Я вижу описание статистического анализа в разделе «Материалы и методы», но в результатах не вижу его применения. Не показано, какие комбинации статистически значимо отличаются.
Answer: Thank you very much for the note. The statistics section has been adjusted.

Round 2
Reviewer 2 Report
This version of the manuscript has been significantly corrected by the authors and at the moment I have no significant objections and I can recommend the text for publication
Author Response
Author responses to reviewer (2) comments
(life-2474325)
This version of the manuscript has been significantly corrected by the authors and at the moment I have no significant objections and I can recommend the text for publication
Answer: Thank you for your valuable comments that helped to significantly improve the quality of presentation of the results of our work for the world community. Thank you!

Reviewer 3 Report
I do not see that the authors have added the results of any statistical tests for comparative analysis. The only statistics presented in the paper are for correlation analysis.
Author Response
Author responses to reviewer (3) comments
(life-2474325)
I do not see that the authors have added the results of any statistical tests for comparative analysis. The only statistics presented in the paper are for correlation analysis.
Answer. Thank you very much for your note. Text have been corrected:
Statistical data processing was carried out using the Statistica 12.0 package. Statistical data (average values, variance) were determined, and the reliability of various samples was established using variance analysis (Student’s t-test) and confidence interval (95%).
Confidence intervals are drawn on each figure (figure 1and 2), but, however, they are not visible everywhere because of the same scale of the X and Y axes (the same because 3 types of soils are compared). The confidence interval is especially noticeable in Haplic Cambisols.
Table 2 presents the calculated values according to the data that have already been processed and are presented in Fig. 1 and 2, so statistical processing indicators are not shown.
